# Effects of Voids on Thermal Fatigue Reliability of Solder Joints on Inner Rings in Ball Grid Array Packaging by Finite Element Analysis

**DOI:** 10.3390/mi14030588

**Published:** 2023-02-28

**Authors:** Xingwang Hu, Li Liu, Sheng Liu, Meng Ruan, Zhiwen Chen

**Affiliations:** 1The Institute of Technological Sciences of Wuhan University, Wuhan University, Wuhan 430072, China; 2School of Materials Science and Engineering, Wuhan University of Technology, Wuhan 430072, China; 3Tsinghua Innovation Center in Zhuhai, Zhuhai 519087, China

**Keywords:** ball grid array, temperature cycling test, thermal fatigue reliability, solder joint, macro voids

## Abstract

Under alternating temperatures, the fatigue failure of solder balls caused by the mismatch of the thermal expansion coefficient is a key problem in a Ball Grid Array (BGA). However, the combined effects of the solder ball location and the size of voids within it can seriously affect the thermal fatigue reliability of BGA solder balls, which can be easily ignored by researchers. Firstly, the thermal fatigue reliability of the board-level solder balls was evaluated by a temperature cycling test of the BGA package at −20 °C–+125 °C. The experimental results showed that the thermal fatigue reliability of the outer ring’s solder joint was lower than that of the inner ring. Secondly, the reliability of the solder balls in the BGA package was studied under the same thermal cycling condition based on finite element analysis (FEA). The influences of voids on fatigue life were investigated. Generally, a linear correlation between the void content and the fatigue life of the inner rings’ solder balls could be identified with a gradually smoothed relationship for solder balls closer to the center. In addition, when the size of the void exceeded a critical volume, the inner ring’s solder ball with the void would fail before the outermost ring. The results of FEA showed that the critical void volume ratio from the second to fifth ring increased from 10.5% to 42.3%. This study provides a valuable reference for the influence of voids on the thermal fatigue reliability of BGA solder balls.

## 1. Introduction

With the continuous miniaturization of electronic products, electronic packaging is also developing in the direction of microscale, high density, and fine pitch. This will lead directly to the miniaturization of solder balls and a series of quality control, durability, and reliability problems [1]. Much research shows that solder balls are the weakest link in the electronic packaging system [2,3]. Due to the changes in ambient temperature and Joule heat caused by current interruption, solder balls in electronic packaging systems normally undergo temperature cycling and the subsequent thermal fatigue load by the mismatch of CTE (Coefficient of Thermal Expansion) during long-term service [4].

There are many factors affecting the thermal fatigue life of solder joints, and the void is one of the most important ones. Voids are very common manufacturing defects in solder joints, and the causes are variable [5]. Macro voids produced by the volatile chemicals of the flux during reflowing are the most harmful, typically 4 to 12 mils (100 to 300 μm) in diameter [6]. It has been found that macro voids will lead to increased resistance and decreased conductivity of the solder ball [7]. Moreover, the thermal stress of a solder ball with macro voids will increase when it is subjected to a temperature cycling load, thereby reducing the thermal-mechanical reliability of the solder ball [8].

The influences of voids on the reliability of solder balls have been reported from several different perspectives in the literature and standards. IPC-7095-C reported the type and location of voids in BGA [9]. IPC-A 610E defined the standard for void content in BGA solder balls, which requires that voids in X-ray images not exceed 30% of the area of a single solder ball [10]. Many works of FEA showed that the size of a void and its position inside the solder joint seriously affected the thermo-mechanical fatigue reliability of BGA. It was found that when the macro void was close to the boundary of the solder ball, its influence on fatigue reliability became greater [11]. With an increase in void size, the sensitivity of durability to void location decreased gradually [12]. However, a void is not always harmful, especially a void far away from the corner of solder joints. It was found that there was no clear dependence between the thermal conductivity of solder joints and void contents [7]. In addition, there are some contradictory results in the exploration of the different effects of multiple small voids and one large void under the same projected area [13,14]. This difference may be caused by the different positions of voids inside the solder joints. Previous studies generally focused on the size of the void and its position inside a single solder ball while ignoring the position of the solder ball itself in the BGA package. It has been reported that a void can lead to the failure of solder balls on the inner ring prior to failures of the outermost ones [15]. Therefore, the location of failed solder balls in BGA is actually a combined result of void size and increased rate of accumulated inelastic strain, which should be accounted for simultaneously.

In this study, the BGA board-level temperature cycling tests at −20 °C–+125 °C were carried out to evaluate the thermal fatigue reliability of solder joints on the four outer rings. BGA FEM (Finite Element Method) models with the same dimensions were also established to investigate the effects of voids on thermal fatigue life. On this basis, voids with progressively increasing diameters were introduced within solder balls at typical locations to investigate the influences of voids on the thermal fatigue reliability of BGA. The critical void volume ratio was further determined for solder joints located at different inner rings of BGA, which was also the minimum void volume ratio that made the inner solder joint fail prior to the outermost ones. This study can provide a valuable reference for design and process optimization in BGA packaging.

## 2. Experimental Procedure and Finite Element Simulation

### BGA Temperature Cycling Test

As shown in Figure 1, 30 Sn96.5Ag3.0Cu0.5(SAC305) BGA chips with daisy chain were stored in a high and low-temperature test chamber with a temperature range of −20 °C–+150 °C. During the thermal fatigue test, the electrical signals of the four rings’ solder joints, starting from the outermost, were monitored independently and in situ to track the failure of each ring.

The temperature profile is shown in Figure 1b. Thermal cycling was performed between −20 °C and +125 °C. The dwell times were 15 min at −20 °C and +125 °C, and it required 40 min for heating and 60 min for cooling in a cycle. That made a total cycle time of 130 min. After temperature cycling tests for 2325 h, an X-ray analysis was performed to examine all samples. Then a dye and pry test was carried out, and the fatigue fracture of the solder joints was observed by SEM.

## 3. Finite Element Analysis

This section may be divided by subheadings and should provide a concise and precise description of the experimental results, their interpretation, as well as the experimental conclusions that can be drawn.

FEA models of BGA were built to explore the influence of voids on the thermal fatigue reliability of solder balls on different rings. The model consisted of the PCB, SAC305 solder balls, BT substrate, die, and EMC (Epoxy Molding Compound), as shown in Figure 2 and Table 1. The 1/4 model was meshed with 214722 hexahedral 20-node SOLID 186 elements. The SOLID 186 element exhibits quadratic displacement behavior and can be used in calculations for plasticity, hyperelasticity, creep, stress stiffening, large deflection, and large strain deformation.

The material properties used for the FEA simulation are presented in Table 2. The viscoplasticity of SAC305 solder balls was defined by the Anand model [16], as listed in Table 3. All the materials were modeled as linear elastic except the solder. To simplify the simulation, several hypotheses were also included in the model: (1) the temperature distribution inside the BGA package was uniform without gradience; and (2) the volume of IMC and the Cu pad were very small compared with the volume of the solder ball, so they were not considered in this study. The accuracy of the simulation can be verified with the results of BGA thermal cycling tests.

Macro voids with different diameters were then added inside the diagonal solder balls of the second to fifth rings, as listed in Table 4 and Figure 2. It should be noted that each simulation model only added a macro void inside one solder ball, such as the model in Figure 2c. The distance between the center of the solder ball and the void was 0.05 mm. By adjusting the diameter of the void, the effects of the volume ratio of the void on the fatigue life of the solder ball can be systematically studied.

## 4. Results and Discussion

### 4.1. Temperature Cycling Test Results

Figure 3a shows the typical output current curves of failed and non-failed samples. A stable current signal could always be received from the non-failed samples. For signals from failed samples, the failure can be identified at the first drop from over 10 mA to 0 mA, which was caused by the open circuit. The following spikes in the curve can be attributed to occasional contacts at the failed sites by the vibration of the chamber during the test. Thermal fatigue results are summarized in Figure 3b. A total of 13 rings’ solder joint failures were monitored in the temperature cycling test.

As can be seen from Figure 3b, the mean fatigue life of the solder balls decreased with the increase in distance between the solder balls and the geometric center of the BT substrate. Correspondingly, the number of failed solder balls increases as this distance increases. The innermost solder balls all survived after 2325 temperature cycles. The reason for this difference is that the amount of inelastic deformation by the mismatch of CTE is proportional to the distance from the solder ball to the geometric center of the substrate.

Since only 10.8% of the samples failed the tests, the right censored Weibull distribution [18] was used to fit the failure data. According to the calculation, the shape parameter, β, and characteristic life, η, of the Weibull distribution in the cycles-to-failure (CTF) are 3.67 and 4184 cycles for the temperature cycling test, respectively.

An X-ray was carried out on all 30 BGA board-level samples after the temperature cycling test. The three solder balls marked in Figure 4b had typical open circuit failures, which were located in the outermost two rings of all solder balls. No failures or defects could be identified in solder balls closer to the geometric center. The void area ratio was calculated based on the X-ray projection images of BGA samples before and after the temperature cycle tests, as shown in Figure 4c. It was found that the mean void area ratio decreases with the drop in distance between the solder balls and the geometric center. After the thermal cycle test of 5037 h, the area of void in the failed solder joint of the outermost ring was still smaller than that allowed by IPC-A 610E. Although voids were also found in the solder balls of the fourth and fifth rings, there were no failed solder joints due to smaller voids.

The results of the dye and pry test are shown in Figure 5. The red area in Figure 5a was the disconnected area caused by thermal fatigue, which has been stained by red ink. The white area was still connected after the thermal fatigue test, so it was not stained by red ink. Compared with Figure 5a,b, it can be seen that the disconnection area is mainly composed of A, B, and C, and their enlarged images are shown in Figure 5c. Similar lamellar morphology can be seen in Figure 5c(A,C) with obvious cracks between them. Area B is porous and granular. EDS results show that area B contains 70 wt.% carbon, which is caused by red ink. Element distribution scanning was carried out for the whole section of the solder joint, and component analysis was carried out for the local parts. The results show that there was no IMC on the fracture surface, which indicates that the fracture occurred near the solder/IMC interface but that no crack passed through the IMC layer.

### 4.2. Finite Element Analysis Results

To evaluate the effects of voids on the fatigue life of solder joints, a BGA model with no voids was first investigated (case 1). Figure 6a is the equivalent von Mises stress distribution of the outermost solder ball after five cycles of temperature cycling loading. The maximum equivalent stress and strain were located at the solder/chip interface at the outermost corner. This solder ball was the one farthest from the geometric center of the substrate and is often referred to as the “dangerous solder ball”. It was found that the “dangerous solder ball” is a weak link, which generally leads to the first failure of the packaging module [19].

Figure 7 shows the evolution of the equivalent von Mises stress and accumulated equivalent plastic strain of four outer rings’ solder balls under the temperature cycling load. The equivalent stress changes periodically with time. However, the amplitude of the equivalent stress dropped with a lower distance to the geometric center. Figure 7b shows that the accumulated plastic strain generally increased as the test continued with the highest rate at the outermost ring. This will also lead to the final failure of solder balls.

A strain-based Engelmaier-modified Coffin-Manson fatigue life prediction model [20] was used to calculate the fatigue life of solder balls in the BGA package. For lead-free solder balls of SAC305, the formula for calculating the fatigue life *N_f_* is as follows:(1)Nf=12Δγ2εf−1c
where Δγ is the equivalent inelastic shear strain range, Δγ = √3Δε; Δε is the equivalent inelastic total strain range; ε*_f_* is the fatigue toughness coefficient, ε*_f_* = 0.48; and *c* is the fatigue toughness index, which is related to the temperature and frequency of the temperature cycle.
(2)c=−0.39−9.3×10−4Tsj+1.91×10−2ln1+100tD
where, *T_sj_* is the average temperature of the temperature cycle (°C), and *t_D_* is the half-cycle dwell time in minutes. As can be seen from Figure 8, the equivalent plastic strain accumulated in a single cycle tends to be stable after the first cycle. Therefore, it is normally recommended to use the accumulated equivalent plastic strain in the fifth cycle to calculate the fatigue life of solder balls.

Based on the accumulated equivalent plastic strain in Figure 7b and Equations (1) and (2), the fatigue life of the solder balls was calculated and listed in Table 5. The fatigue life decreased with a higher distance to the geometric center. The fatigue life of the “dangerous solder ball” located at the outermost corner was 4014 cycles from FEA, which was 4.1% different from the characteristic life of a solder ball obtained by the BGA thermal cycling test. This shows that the results of the simulation and experiment were consistent.

To further investigate the influences of voids on the thermal fatigue reliability of BGA, voids with progressively increasing diameters were introduced within solder balls at typical locations (case 2–21 in Table 4). The typical results of the comparison of the stress and strain distribution between case 2 with a void and case 1 without a void are shown in Table 6. It can be seen that the effects of the void in the solder joint of the second ring on the equivalent stress and strain of the outermost rings in case 2 were extremely small when comparing the stress/strain distribution of the outermost solder balls in case 1 and case 2. When comparing the solder ball at the second ring in case 2 with the solder balls at the outermost and second rings in case 1, it showed that the macro void could not affect the locations of maxima but that the stress and strain concentrations at the solder/chip interface deteriorated severely with the introduction of voids. This leads to the increase of the maximum stress/strain in the solder ball with a void at the second ring in case 2, which will likely exceed the maxima of the outermost solder ball in case 1.

Figure 9 shows the accumulated equivalent plastic strain of solder joints on the outermost ring in case 1 and the second ring in cases 2–6. The accumulated equivalent plastic strain of solder balls on the second ring will gradually increase with a larger void volume and even exceed that of the outermost ones without voids. In this process, a critical void volume ratio can be determined, which makes the accumulated plastic strain of the solder ball on the second ring equivalent to the outermost ones, such as in case 1 and case 4 in Figure 9. According to formulas 1 and 2, the fatigue life of the two was also equivalent.

Based on the simulations listed in Table 4, the similar effects of the void volume on the accumulated plastic strain can also be identified in solder balls at other inner rings, as summarized in Figure 10a. It can be found that for solder balls at different inner rings, the accumulated plastic strain generally increased linearly with the void volume from less than that at the outermost solder ball in case 1 to surpassing it. Therefore, there is always a critical void volume for solder balls at the inner rings, at which the accumulated plastic strain in solder balls at inner rings was equivalent to that in the outermost solder ball (the interception points in Figure 10a).

Based on Equations (1) and (2), the thermal fatigue life of solder balls from simulations in Table 4 is also evaluated and summarized in Figure 10b. In contrast to accumulated plastic strain, the increase in void volume led to a significant drop in fatigue life. The slopes of the fitted curves gradually decreased with a lower distance to the geometric center. This means that the fatigue life of solder joints farther from the geometric center tends to be more sensitive to void sizes, which can also be attributed to the fact that the solder joint of the outer ring withstands greater inelastic strain from the mismatch of CTE.

Furthermore, from Figure 11, the critical void volume ratio dropped linearly with a higher distance to the geometric center. It can be expressed as:(3)y=−10.83x+80.03
where *x* is the distance from the solder ball to the center of the substrate, and *y* is the critical void volume ratio. The solder ball closer to the interior can withstand larger voids. Among them, for the solder joints of the second and third rings, when the void volume ratio exceeded 10.5% and 19.6%, respectively, their increased rate of accumulated inelastic strain during thermal cycling would exceed the outermost ones without voids, so that failures occurred first. This is also the reason why partial failures were also monitored in the second and third rings’ solder joints in the thermal cycling tests. However, the critical void volume ratio in the fourth and fifth rings’ solder joints reached 30.1% and 42.3%, respectively. Such a large void can seldom be found in solder balls [21,22]. This is also the reason why no thermal fatigue failure can be found within the fourth ring in the temperature cycling test. The results of this study show that the solder balls on the inner ring have a stronger tolerance to voids. However, the requirements of the current standards on the voids of solder joints are half-baked and oversimplified. For example, IPC-A 610E requires that voids in X-ray images not exceed 30% of the area of a single solder ball. In addition, from the perspective of practical application, the fatigue life of the BGA module and the tolerance of deviation in the process parameter can be significantly improved by adding several rings of solder joints outside the solder joints of key functions.

## 5. Conclusions

In this paper, thermal cycling tests were carried out on BGA samples with a temperature range of −20 °C to +125 °C for 2325 cycles (5037 h). FEA models of BGA with the same dimensions were established to further investigate the influences of macro voids on the thermal fatigue reliability of solder balls at representative locations. The conclusions are as follows.

A total of 13 rings’ solder joints failed in the temperature cycling test. The characteristic life was 4184 cycles after fitting with the Weibull distribution theory. According to the Coffin-Manson model, the fatigue life of solder balls was predicted based on FEA modeling, and it was reasonably consistent with the experimental results.In models with macro voids in the solder balls at typical positions, it was found that the stress and strain concentrations at the solder/chip interface deteriorated severely. The accumulated equivalent plastic strain of solder balls will gradually increase with higher void volume and even exceed that of the outermost ones without voids. For solder balls with voids, it was found that the critical void volume ratio dropped at a higher distance to the geometric center. Meanwhile, the fatigue life of solder joints farther from the geometric center tends to be more sensitive to void sizes.The effect of a single macro void on the thermal fatigue reliability of solder balls was investigated, which provided a valuable reference for design and process optimization in BGA packaging.

In the future, it is necessary to quantitatively evaluate the effect of multiple small voids and void spacing on the thermal fatigue reliability of solder joints at different positions in BGA through reliability testing. Therefore, the standards and related process window for BGA can be further improved.

## Figures and Tables

**Figure 1 micromachines-14-00588-f001:**
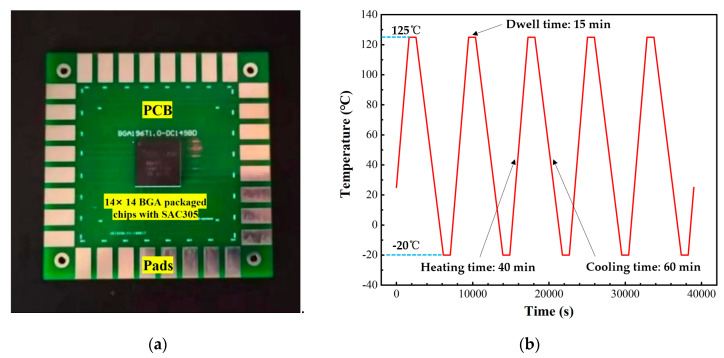
(**a**) PCB test board in thermal cycling test. (**b**) Temperature profile.

**Figure 2 micromachines-14-00588-f002:**
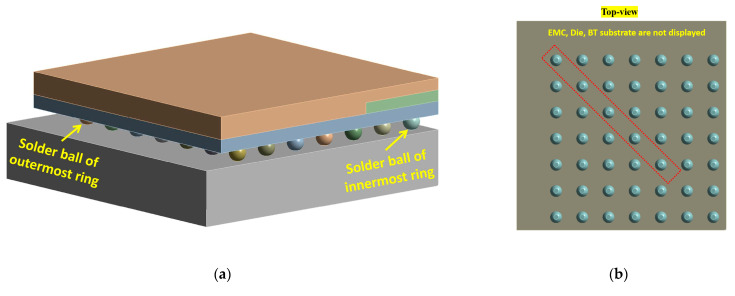
(**a**) Locations of critical solder balls. (**b**) Solder ball array of 1/4 BGA model. (**c**) Macro void in solder ball of 1/4 BGA model.

**Figure 3 micromachines-14-00588-f003:**
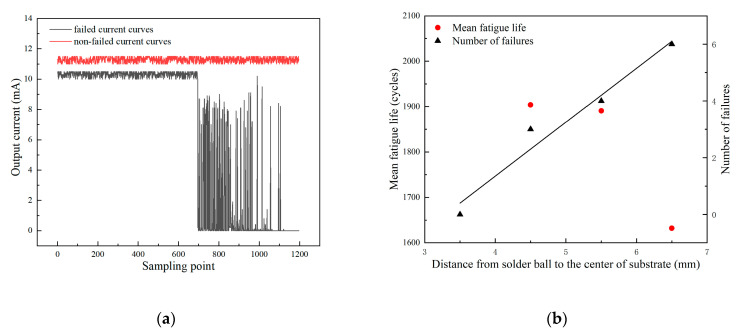
(**a**) Typical failed and non-failed current curves. (**b**) Results of thermal fatigue test.

**Figure 4 micromachines-14-00588-f004:**
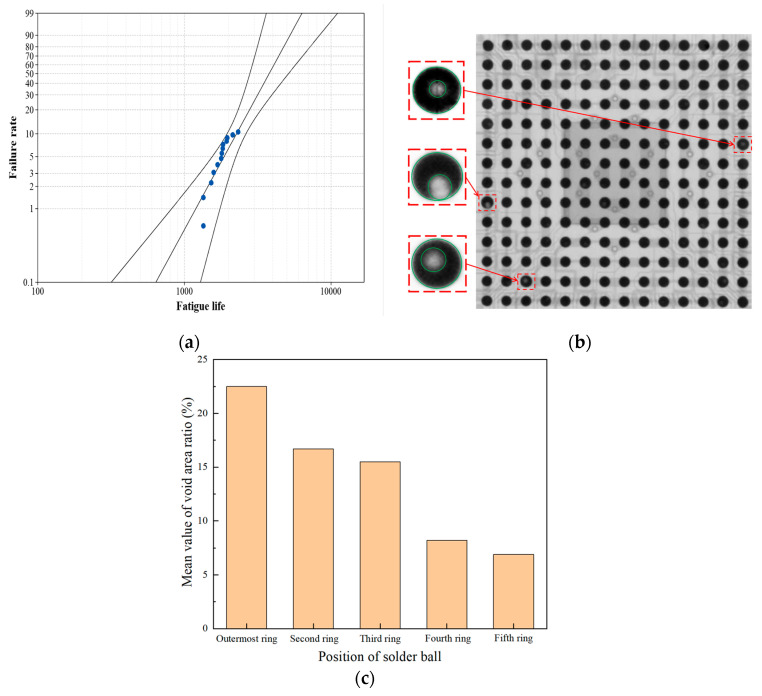
(**a**) Weibull distribution of failure life. (**b**) X-ray flaw detection image after temperature cycling test, (**c**) Mean void area ratio in X-ray images.

**Figure 5 micromachines-14-00588-f005:**
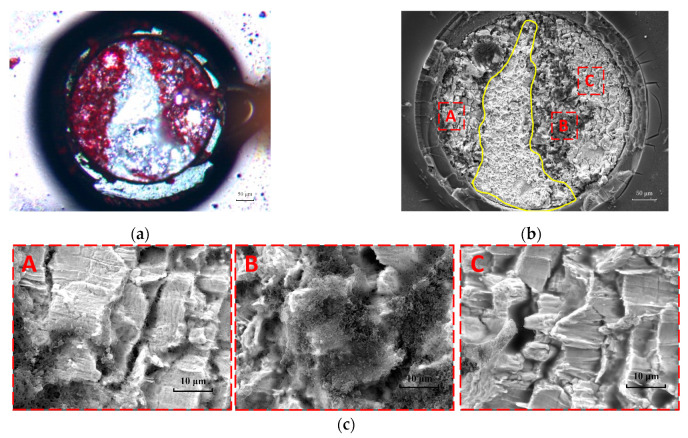
(**a**) Optical microscopy image of solder joint fracture. (**b**) SEM image after dye and pry test of BGA. (**c**) Enlarged images of disconnect area in Figure (**b**). [(**A**) Lamellar grains (**B**) porous grains (**C**) Lamellar grains.].

**Figure 6 micromachines-14-00588-f006:**
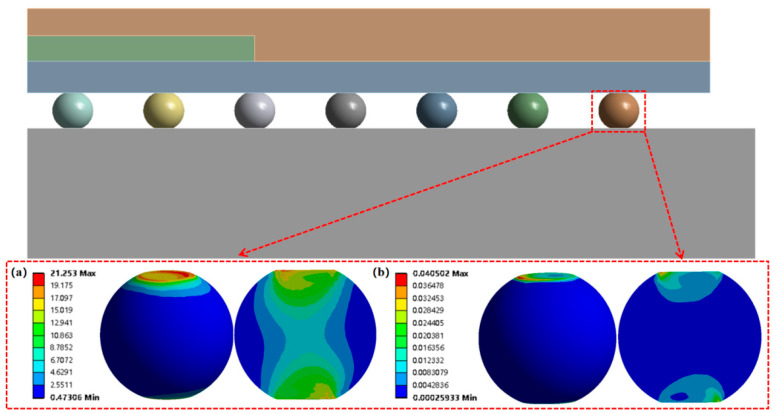
(**a**) Distribution of equivalent von Mises stress. (**b**) Distribution of accumulated equivalent plastic strain of “dangerous solder ball” in case 1.

**Figure 7 micromachines-14-00588-f007:**
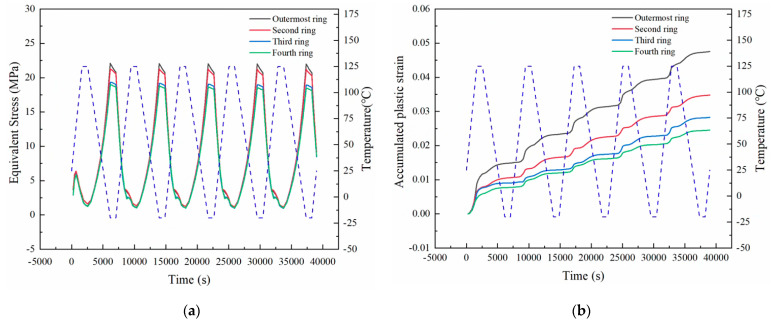
(**a**) Equivalent stress. (**b**) Accumulated equivalent plastic strain of four outer rings’ solder balls.

**Figure 8 micromachines-14-00588-f008:**
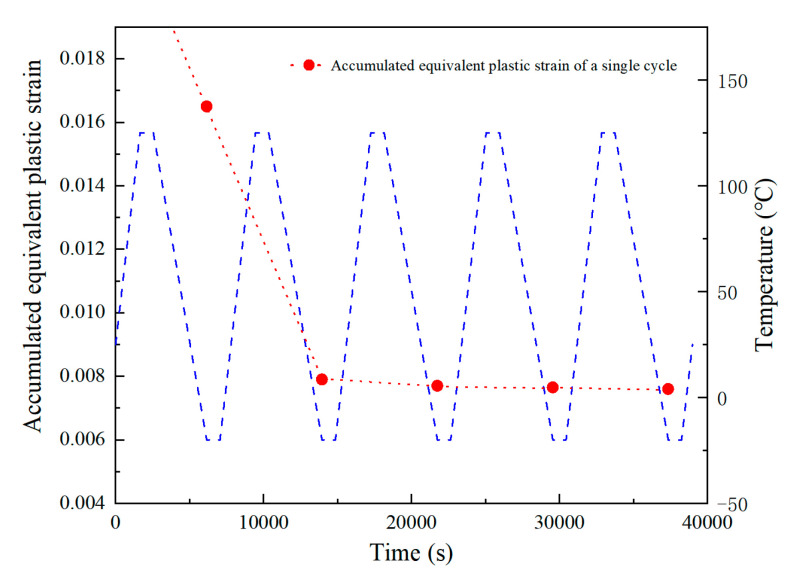
Accumulated equivalent plastic strain by a single cycle during temperature cycling of the “dangerous solder ball” in case 1.

**Figure 9 micromachines-14-00588-f009:**
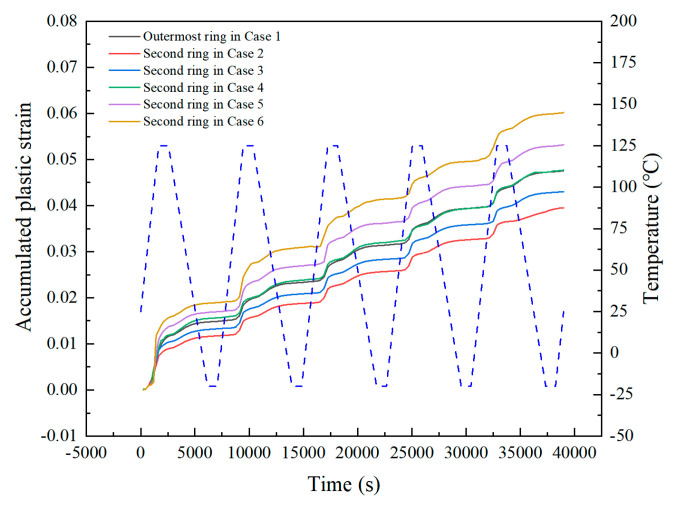
Comparison of accumulated equivalent plastic strain of solder joints on outermost ring in case 1 and second ring in cases 2–6.

**Figure 10 micromachines-14-00588-f010:**
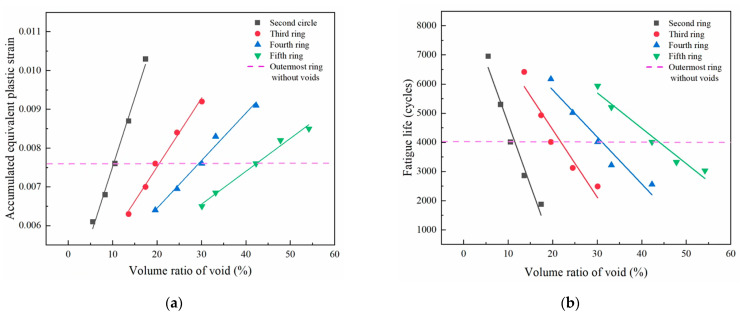
The relationship between (**a**) void content and accumulated equivalent plastic strain in a single circle and (**b**) void content and fatigue life of solder balls.

**Figure 11 micromachines-14-00588-f011:**
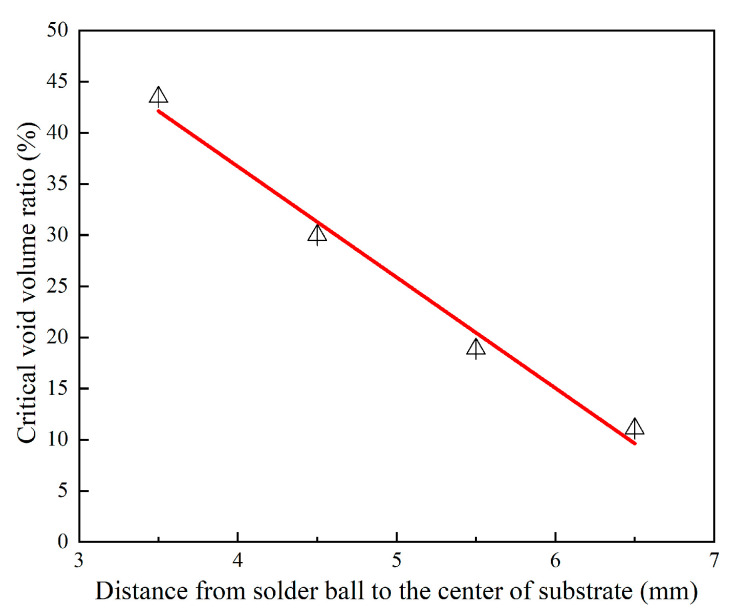
The relationship between critical void volume ratio and location of solder ball.

**Table 1 micromachines-14-00588-t001:** Dimensions of each component in the BGA package.

Components	Sizes of Solder Balls	Substrate	Die	PCB	EMC	Solder Ball Pitch
**Sizes (mm)**	φ0.45 × 0.41	15 × 15 × 0.36	5 × 5 × 0.3	75 × 75 × 1.5	15 × 15 × 0.61	1

**Table 2 micromachines-14-00588-t002:** Material properties of components in the BGA package [17].

Components	Density (kg/m^3^)	Elastic Modulus (MPa)	Poisson’s Ratio	Coefficient of Thermal Expansion (10^−6^/K)
Substrate	1660	(xy) 17,890	(xy) 0.11	(xy) 14.5
		(z) 7846	(z) 0.39	(z) 67.2
PCB	9920	(xy) 19,303	(xy) 0.11	(xy) 14.5
		(z) 8476	(z) 0.39	(z) 67.2
Solder ball	7400	41,600	0.35	24
Die	2330	163,000	0.28	2.5
EMC	1660	23,520	0.30	15

**Table 3 micromachines-14-00588-t003:** Anand Parameters of Sn3.0Ag0.5Cu solder [18].

S_0_ (MPa)	Q/R (K)	A (1/s)	h_0_ (MPa)	m	s (MPa)	n	a	ξ
5.1	3468	0.00093	49,000	0.065	16.4	0.078	4	3.2

**Table 4 micromachines-14-00588-t004:** BGA models with macro voids at different locations.

Model Type	Position of Solder Ball with Macro void	Diameter of Macro Void	Volume Ratio of Macro Void
Case 1	None	0	0%
Case 2–6	Second ring	175~250 μm	5.5~17.4%
Case 7–11	Third ring	230~300 μm	13.6~30.1%
Case 12–16	Fourth ring	260~335 μm	19.6~42.3%
Case 17–21	Fifth ring	300~365 μm	30.1~54.2%

**Table 5 micromachines-14-00588-t005:** Fatigue life of solder joint on outer four rings.

Position of Solder Joint	Outermost Ring	Second Ring	Third Ring	Fourth Ring
Accumulated equivalent plastic strain Δε during the fifth cycle	0.0076	0.0058	0.0051	0.0039
Fatigue life (cycles)	4014	7890	10,883	21,282

**Table 6 micromachines-14-00588-t006:** Effect of the void on distribution of equivalent stress/strain of solder balls.

Stress/Strain	Equivalent Stress	Equivalent Plastic Strain
**Model Number**	**Case 1**	
Outermost ring	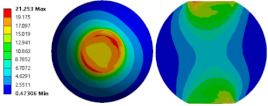	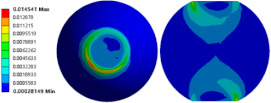
Second ring	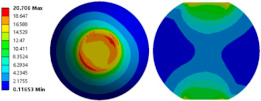	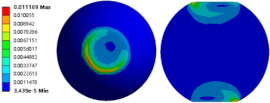
**Model Number**	**Case 2**	
Outermost ring	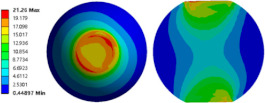	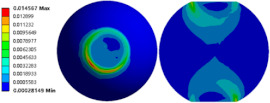
Second ring	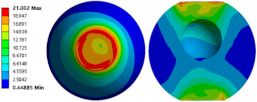	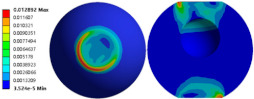

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
