# Peer review of "Effects of Voids on Thermal Fatigue Reliability of Solder Joints on Inner Rings in Ball Grid Array Packaging by Finite Element Analysis"

_micromachines, 2023, doi:10.3390/mi14030588_

Round 1
Reviewer 1 Report
Voids in solder joints are serious concerns in reliability of electronic devices. And they are also one of the most common defects in production. This manuscript presented some interesting results about the tolerable void ratio in BGA, which can provide some new and valuable information to the community. But there are also some problems to be corrected before it can be published.
1. The layout of this manuscript should be checked and improved, such as case 1 and case 2 in table 6, table 1 and fig 4.
2. More information about the model should be presented in section 3, such as the layout and number of solder balls in the model.
3. Where are the solder balls with void located exactly in the models?
4. Is it possible to quantify the correlation between critical void ratio and location of solder balls?
5. There are several standards about voids in BGA for the quality control in production, such as IPC standards. Some analysis and discussions should be presented regarding results in this work and the relevant standards.
6. Full names should be given for abbreviations where they first appeared, such CTE, FEA, FEM, EMC and others.
7. The manuscript should be checked more carefully for language and format issues, such as “min”, “5037h”.
Reviewer 2 Report
Effects of Voids on Thermal Fatigue Reliability of Solder Joints on Inner Rings in BGA Packaging
This manuscript describes an evaluation of the thermal fatigue reliability of solder joints on the four outer rings of a BGA. In addition, it was used to study the effects of voids on thermal fatigue life. It is a very interesting topic to study.
Title:
1. The title is unclear when it comes to experimental, or simulation works. change the title, for example: “Effects of Voids on Thermal Fatigue Reliability of Solder Joints on Inner Rings in Ball Grid Array Packaging by Finite Element Analysis”.
2. Titles should not contain abbreviations. It's fine for popular acronyms such as AIDS, USA, etc. Some readers are not familiar with specific abbreviations. Example: BGA.
Abstract:
1. Abstracts generally consist of four items: introduction to the work, the method that was used, main findings and novelty/conclusion of the work.
2. Abstract is too long.
3. too many words “In this paper”, in this work, this study,
4. Line 20: What is the different between “finite element method” and “finite element analysis”?
5. A very little information is provided about FEA, but the whole work is centered around simulation.
Introduction:
1. Introduction should at least 3 components/paragraphs:
a. introduction to the field
b. problem statement
c. objective and method/approch
2. L31-42: This paragraph contains very general information about solder, PCBs, etc. It needs to be reduced.
3. No literature about fatigue or void analysis using FEA.
4. L71-81: Shorten the paragraph, focus on "objective and method/approach"
5. What is CTE?
Experimental:
1. L89: There is no information about the temperature taken for each BGA. How about thermal sensor etc.?
2. Figure 1: Temperature profile is too general, where I was taken?
3. L120: “The macro voids with different diameters were then added inside the solder balls…”. Please provide a picture showing how it was done.
Results and Discussion:
1. Figure 3b: Three points of data are not enough to plot liner lines for fatigue life.
2. L149: “The void area percentage in the X-ray images”. It would be helpful if you could provide more details about this. Even FESEM images can be helpful in validating void and IMC formations.
3. L164: “cracks tend to initiate from…”. How does this crack look on a real sample? Do you have a picture?
4. L195: “It shows that the built FEM models were in reasonable consistency with experimental results.” This conclusion needs more evidence (physical evidence).
Conclusion
1. Need to be answered your objectives, problem statements and outlook form this work.
2. It should be a paragraph or a list of bullet points.
3. Too long for current conclusion.
4. L252-256: Rephrased into two sentences.
5. L157-266: The bullets 1-3 should answer your objective and problem statement. make I concise.
6. L267-274: Describe the outlook/novelty of this study in two sentences.
Refs
1. References must be within 5 years (2019-2023)
2. In-text citations and references should be consistent throughout the manuscript.
Round 2
Reviewer 2 Report
Authors have provided me with a comprehensive response, and I am satisfied with it.